# Histone Methyltransferase *Ss*Dim5 Regulates Fungal Virulence through H3K9 Trimethylation in *Sclerotinia sclerotiorum*

**DOI:** 10.3390/jof10040271

**Published:** 2024-04-06

**Authors:** Lei Qin, Xin Gong, Jieying Nong, Xianyu Tang, Kan Cui, Yan Zhao, Shitou Xia

**Affiliations:** 1Hunan Provincial Key Laboratory of Phytohormones and Growth Development, Hunan Agricultural University, Changsha 410128, China; leiqin2020@stu.hunau.edu.cn (L.Q.); xingong@stu.hunau.edu.cn (X.G.); njy@stu.hunau.edu.cn (J.N.); chuxuantingnasha@stu.hunau.edu.cn (X.T.); zhaoyan3@venusgroup.com.cn (Y.Z.); 2Institute of Plant Protection, Hunan Academy of Agricultural Sciences, Changsha 410125, China; ck0601@stu.hunau.edu.cn

**Keywords:** *S. sclerotiorum*, pathogenicity, histone methylation, mycotoxins, stress

## Abstract

Histone post-translational modification is one of the main mechanisms of epigenetic regulation, which plays a crucial role in the control of gene expression and various biological processes. However, whether or not it affects fungal virulence in *Sclerotinia sclerotiorum* is not clear. In this study, we identified and cloned the histone methyltransferase *Defective in methylation 5* (*Dim5*) in *S. sclerotiorum*, which encodes a protein containing a typical SET domain. *SsDim5* was found to be dynamically expressed during infection. Knockout experiment demonstrated that deletion of *SsDim5* reduced the virulence in *Ssdim5-1/Ssdim5-2* mutant strains, accompanied by a significant decrease in H3K9 trimethylation levels. Transcriptomic analysis further revealed the downregulation of genes associated with mycotoxins biosynthesis in *SsDim5* deletion mutants. Additionally, the absence of *SsDim5* affected the fungus’s response to oxidative and osmotic, as well as cellular integrity. Together, our results indicate that the H3K9 methyltransferase *Ss*Dim5 is essential for H3K9 trimethylation, regulating fungal virulence throug mycotoxins biosynthesis, and the response to environmental stresses in *S. sclerotiorum*.

## 1. Introduction

*Sclerotinia sclerotiorum* is a widespread fungal pathogen that parasitizes various hosts, causing severe diseases in over 600 plant species globally [1,2], which have significantly impact on the yield and quality of important economic crops such as canola and soybeans [3]. To successfully infect and parasitize its hosts, *S. sclerotiorum* has evolved complex and sophisticated infection strategies. Firstly, *S. sclerotiorum* forms appressoria through differentiation of tip hyphae, a multicellular, melanin-rich hyphal penetration structure [4,5], which adhere to and penetrate the host surface, thereby breaking through the host’s physical barrier. Simultaneously, *S. sclerotiorum* secretes a non-host-specific toxin, oxalic acid (OA), which can interfere with the host’s redox environment and pH signaling to promote pathogenicity [6]. In addition, *S. sclerotiorum* can also secrete some secreted proteins through appressoria in the early stage of infection to overcome the plant’s defense response and establish a short biotrophic stage [2]. Notably, *SsCmu1* is significantly upregulated during infection of *Brassica napus* [7], which encodes a secreted cutinase, can inhibit host salicylic acid synthesis, thereby promoting infection [8]. Other putative effectors, such as *Ss*ITL [9], *Ss*CM1 [10], *Ss*CVNH [11], *Ss*v263 [12], and fungal effector proteins containing LysM domains, also exhibit high upregulation in the early infection stages, impacting the virulence of *S. sclerotiorum* [13]. After a brief biotrophic phase, the fungus induces host cell necrosis and cell wall degradation by producing numerous toxins, necrosis-inducing secreted proteins, and cell wall-degrading enzymes [2].

For pathogens, in order to adapt to changing host environments and defenses, infection processes require rapid and subtle adjustments to their gene expression programs [14,15,16]. The regulation of gene expression is achieved at the transcriptional level through various mechanisms, include histone modification, as one of the primary epigenetic regulatory mechanisms, playing a particularly important role in shaping fungal pathogenicity [17]. Histones can undergo various covalent modifications, including methylation, acetylation, phosphorylation, and ubiquitination [18]. Histone methylation primarily occurs on the side chains of lysine and arginine. Lysine can undergo monomethylation, dimethylation, or trimethylation, while arginine may undergo monomethylation, symmetric dimethylation, or asymmetric demethylation [18,19]. Histone lysine methylation, a process orchestrated by histone lysine methyltransferases (HKMTs) with the SET domain, involves the transfer of a methyl group from S-adenosyl-L-methionine (SAM) to lysine residues at the N-terminus of H3 or H4 histones [20]. In eukaryotes, histone methylation is an epigenetic mechanism associated with various functions related to gene regulation or genome stability. H3K9 methyltransferases are responsible for establishing histone H3 lysine 9 methylation (H3K9me), which is closely associated with constitutive heterochromatin and participates in numerous biological processes [21]. For instance, *Schizosaccharomyces pombe* can regulate its mating type, chromosome segregation, and growth development by modulating the methylation levels of H3K9 [22,23]. In *Fusarium proliferatum* and *Fusarium mangiferae*, H3K9 trimethylation not only regulates their growth but also influences the production of secondary metabolites [24,25]. Additionally, pathogenic microorganisms can modulate their virulence by controlling the trimethylation levels of H3K9. For example, *Epichloë festucae* can regulate its virulence through histone H3K9 and H3K36 trimethylation [26]. As a result, histone modifications play a crucial role in the regulation of gene expression, especially during the infection process. Through these modification mechanisms, fungi can flexibly respond to dynamic changes in the host environment.

H3K9 methylation is facilitated by specific proteins, including Clr4 (Cryptic loci regulator 4) in *S. pombe* and Dim5 (Defective in methylation 5) in *Neurospora crassa* [27]. In *Botrytis cinerea*, a plant pathogenic fungus, the absence of the *Dim5* gene results in a significant reduction of H3K9me3, causing the downregulation of pathogenic genes related to host signal perception, host tissue colonization, stress response, toxin synthesis, and host immune response [28]. In the maize pathogen *Fusarium verticillioides*, disruption of *Dim5* significantly reduces H3K9me3 levels, leading to a pronounced decrease in fungal virulence, accompanied by an unexpected increase in osmotic stress tolerance and expression of melanin synthesis genes [29]. In *F. mangiferae*, the absence of the *KMT1* gene significantly impedes the biosynthesis of fumonisin and deoxynivalenol toxins in mango [25]. These indicate that the maintenance of H3K9me3 is crucial for the virulence of pathogenic fungi. However, currently there is no report on the biological function of H3K9me3 in *S. sclerotiorum.*

In this study, we characterized the histone H3 lysine methyltransferase *Ss*Dim5 by exploring its roles in H3 lysine trimethylation, regulation of mycotoxins synthesis, and pathogenicity as well as response to external stress. Our aim was to gain deeper insights into the physiological function of epigenetic regulation in the fungal pathogen *S. sclerotiorum*.

## 2. Materials and Methods

### 2.1. Fungal Strains, Plants, and Culture Conditions

The wild-type (WT) strain 1980 of *S. sclerotiorum* [30] was cultured on Potato Dextrose Agar (PDA), while the knockout mutants *Ssdim5-1/Ssdim5-2* were grown on PDA supplemented with 150 μg/mL hygromycin (Roche, Basel, Switzerland). The genetic complementation strain *SsDim5-C* was cultured on PDA containing 100 μg/mL Geneticin (G418). All of these strains were incubated at 20 °C.

*Nicotiana benthamiana* and *B. napus* (ZS11) plants used for pathogenicity tests were obtained from Hunan Provincial Key Laboratory of Phytohormones and Growth Development and cultivated at 22 °C with a 16-h light/8-h dark photoperiod.

### 2.2. Bioinformatics Analysis of SsDim5

First, the genomic sequence of *BcDim5* (GenBank accession: XM_001550366.1) from *B. cinerea* (Assembly ID: ASM14353v4) was obtained by accessing the NCBI database (http://www.ncbi.nlm.nih.gov/, accessed on 30 May 2023). BlastP analysis was then conducted to identify its orthologs in *S. sclerotiorum* (Assembly ID: ASM14694v2) and other species. Subsequently, multiple sequence alignment was performed using DNAMAN 6.0 (Lynnon BioSoft, Quebec, QC, Canada). Finally, a phylogenetic tree was constructed using the Maximum Likelihood method with MEGA 6.0 software [31].

### 2.3. Gene Knockout and Genetic Complementation of SsDim5

Refer to the previous method [32], sequences of the *SsDim5* gene were amplified using genomic DNA from WT strain 1980 as a template. Subsequently, these sequences were fused with the left and right portions of the hygromycin expression cassette, generating the gene knockout fragment. The knockout fragment was then introduced into WT protoplasts using PEG-mediated protoplast transformation [33]. Selection was performed by screening on PDA medium containing 150 mg/L hygromycin, resulting in the putative knockout transformants of *SsDim5*. Finally, through successive sub-culturing of mycelial tips, pure knockout strains were obtained and confirmed by PCR and qRT-PCR.

Simultaneously, using genomic DNA from the WT as a template, the sequence containing the full-length *SsDim5* gene, including its native promoter, was amplified. After digestion with *Kpn*I and *Eco*RI enzymes, this fragment was ligated into the linearized pCH-NEO1 vector. The vector containing the *SsDim5* gene fragment was introduced into the knockout mutant *Ssdim5-1* through *Agrobacterium tumefaciens*-mediated transformation, thereby obtaining a genetically complemented strain.

### 2.4. DNA Extraction, RNA Extraction and cDNA Synthesis

Fresh mycelia were inoculated onto PDA plates covered with glass paper. Following a two-day incubation, the mycelia were harvested, rapidly frozen in liquid nitrogen, and subsequently pulverized into powder. Genomic DNA extraction was performed using the cetyltrimethylammonium bromide (CTAB) method, as described by Allen et al. [34].

Fresh mycelia were inoculated onto PDA plates covered with glass paper, and hyphae and sclerotia samples were collected after one to seven days of growth. Then, fresh mycelia were ground into a homogenate in PDB, with OD600 = 1.0, and evenly spread onto *B. napus* leaves. Samples were collected at 0, 3, 6, 9, 12, 24 and 48 h post-inoculation respectively. A commercial RNA extraction kit (AG21019, Accurate Biotechnology (Hunan) Co., Ltd., Changsha, China) was utilized for RNA extraction. Following the manufacturer’s guidelines, first-strand cDNA synthesis was conducted using the Evo M-MLV reverse transcription kit (AG11705, Accurate Biotechnology (Hunan) Co., Ltd., Changsha, China).

### 2.5. Quantitative Real-Time PCR (qRT-PCR) Analysis

The qRT-PCR experiments were conducted using the StepOne™ Real-Time PCR System and the SYBR^®^ Green Premix Pro Taq HS qPCR Kit II (AG11702, Accurate Biotechnology (Hunan) Co., Ltd., Changsha, China). The PCR program consisted of 40 cycles, including an initial denaturation at 94 °C for 2 min, denaturation at 94 °C for 15 s, and annealing at 58 °C for 1 min. *SsTubulin1* (*SS1G_04652*) was employed as the reference gene. All primers can be found in Appendix A. The analysis of relative gene expression levels utilized the 2^^(−ΔΔCT)^ method [35].

### 2.6. Inoculation and Virulence Determination

Inoculation was performed following previously established protocols [36]. Mycelial plugs obtained from the actively growing colony edge were used for inoculating *N. benthamiana* and *B. napus* leaves (diameter 6 mm). Following inoculation, the leaves were cultured at 22 °C with a relative humidity of 95–100%. Photographic documentation was carried out 24 h post-infection.

### 2.7. Appressorium Observation and Oxalic Acid Analysis

Mycelial plugs (diameter 6 mm) were obtained from the actively growing edges. The mycelia were embedded in agar blocks and placed on glass slides. After 16 h of incubation, the morphology and quantity of adherent cells were observed. Additionally, Mycelial plugs (diameter 1 mm) were inoculated onto the epidermis of onion epidermis. After 16 h of invasion, the onion epidermis was stained in a 0.5% trypan blue solution for 30 min. Subsequently, bleaching solution (ethanol:acetic acid:glycerol = 3:1:1) was used for decolorization. Samples were observed and photographed using an optical microscope (Axio Imager 2, ZEISS, Oberkochen, Germany).

Furthermore, mycelial plugs (diameter 6 mm) were inoculated on PDA medium containing 100 μg/mL bromophenol blue to assess oxalic acid secretion.

### 2.8. Western Blot Analysis of H3K9 Trimethylates

The mycelia were collected from PDA plates covered with glass paper membrane, approximately 500 mg in weight. After rapid freezing in liquid nitrogen and grinding, the mycelial powder was added to 100 μL of protein extraction buffer (200 mM pH = 6.8 Tris-HCl, 40% glycerol, 20% β-mercaptoethanol, 8% SDS, 0.4% bromophenol blue) for total protein extraction. The obtained proteins were then separated on a 10% denaturing polyacrylamide gel using sodium dodecyl sulfate-polyacrylamide gel electrophoresis (SDS-PAGE) and subsequently transferred to a nitrocellulose membrane using a Bio-Rad blotting apparatus.

After incubation with the primary antibody, incubation with the secondary antibody followed, and protein detection was carried out using the SuperSignal West Pico PLUS (product number: 34095, Thermo Scientific, Waltham, MA, USA). The antibodies used, along with their sources and dilutions, were as follows: monoclonal antibody against histone H3 (Immunoway, 1:2000, Plano, TX, USA), monoclonal antibody against histone H3K9 (Abcam, 1:2000, Cambridge, UK), and HRP-conjugated Rabbit anti-mouse IgG (1:5000).

### 2.9. Abiotic Stress Response Assay

To assess the response of the *Ssdim5-1/Ssdim-2* mutants to various stresses, we cultured them on PDA media supplemented with 1M NaCl, 1M KCl, 1M sorbitol, 0.005% SDS and 10mM H_2_O_2_, respectively. After incubation for 48 h, we measured the mycelial diameter and calculated the growth inhibition rate using the formula: Growth Inhibition Rate (%) = 100 × (Colony diameter on pure PDA—Colony diameter under different stress conditions) / (Colony diameter on pure PDA).

### 2.10. RNA Sequencing and Data Analysis

Cultures of WT and *Ssdim5-1* mutant mycelia, grown in PDB medium for two days, were harvested for transcriptome sequencing at Biomarker Technologies Corporation. Sequencing was performed on the Illumina NovaSeq platform according to the manufacturer’s instructions, using the Illumina NovaSeq6000 sequencing platform to perform PE150 mode sequencing, and the raw data were modified using a perl script to obtain sequences with Q > 10. Clean_data were mapped to the reference genome using Hisat2 2.0.4. StringTie 2.2.1 was used to splice the cropped copies, gffcompare 0.12.6 was used to compare with the reference genome annotation. Original counts were normalized using fragments per kilobase of transcript per million mapped reads (FPKM), and differential expression analysis was carried out with DESeq2 1.30.1. Genes meeting the criteria of a corrected *p*-value < 0.01 and a fold change ≥2, as determined by DESeq2 analysis, were considered differentially expressed. ClusterProfiler 4.4.4 and topGO 2.48.0 were used for GO and KEGG analysis.

## 3. Results

### 3.1. Identification of S. sclerotiorum Histone H3K9 Methyltransferase

At present, H3K9me3 modification is considered as an epigenetic mark that manipulates gene expression by regulating chromosomal accessibility [37]. To identify the H3K9 methyltransferase in the *S. sclerotiorum* and investigate its biological functions, we utilized the *B. cinerea Dim5* as a blast template and identified the homologous gene *SS1G_01550* in the *S. sclerotiorum* (hereafter referred to as *SsDim5*). Gene structure analysis revealed that *SsDim5* consists of three exons and two introns, encoding a peptide of 308 amino acid residues. Protein BLAST analysis indicated that Dim5 orthologs were widely distributed among species of the *Sclerotinia*, *Monilinia*, *Fusarium*, *Colletotrichum*, *Alternaria*, and *Rutstroemia P*. To analyze the phylogenetic relationships among Dim5 orthologs, we constructed a phylogenetic tree using Dim5 orthologs from the aforementioned species. The results indicated a close evolutionary relationship between *Ss*Dim5 and its ortholog in the *B. cinerea* (Figure 1A). Furthermore, protein structure analysis revealed that Dim5 orthologs from these species all contained the typical domains of H3K9 methyltransferase, including the PreSET domain, SET domain, and PostSET domain. The SET domain can form the AdoMet binding site and catalytic active site of the methyltransferase, and participate in the formation of the hydrophobic structure of DIM5. The PreSET domain can combine with three zinc ions to form a zinc cluster, while the PostSET domain mediates the binding of DIM5 to AdoMet (Figure 1B). This suggests that these orthologs may possess histone modification activity, and that the protein encoded by *SS1G_01550* in *S. sclerotiorum* is an ortholog of the *B. cinerea* Dim5.

### 3.2. Expression Patterns of SsDim5 during Development and Infection Stages

In order to elucidate the regulatory functions of *SsDim5* during the developmental and invasive stages of the *S. sclerotiorum*, quantitative real-time PCR was employed to examine the accumulation of its transcripts at different stages. The expression profile revealed distinct expression patterns of *SsDim5* at various stages. Overall, during the developmental stage of the *S. sclerotiorum*, the expression of *SsDim5* showed an increasing trend, reaching its highest level during the formation and maturation stages of the sclerotia (Figure 2A). Conversely, during different stages of *S. sclerotiorum* infection on *B. napus* leaves, the expression of *SsDim5* was significantly suppressed initially at 9–24 h post-infection, followed by a significant upregulation at 48 h post-infection (Figure 2B). This indicates that *SsDim5* might play a crucial regulatory role in the formation of sclerotia and the infection process on the host.

### 3.3. Generation of SsDim5 Knockout Mutants and Genetic Complementation Strains

To investigate the function of *SsDim5* within the *S. sclerotiorum*, a homologous recombination strategy was employed to generate two independent knockout mutants, *Ssdim5-1* and *Ssdim5-2* (Appendix A). PCR analysis confirmed the absence of the *SsDim5* gene segment in both mutants, replaced by the *HYG* gene segment (Appendix A). The results of qRT-PCR also indicated a lack of *SsDim5* transcript in the knockout mutants (Appendix A). Subsequently, the accuracy of gene complementation in the *Ssdim5-1* mutant was validated by transforming the wild type *SsDim5* gene segment, including a 1500 bp native-promoter, into the mutant strain. PCR and qRT-PCR further confirmed the accuracy of the gene complementation strain, designated as *SsDim5-C* (Appendix A).

WT, two *SsDim5* knockout mutants (*Ssdim5-1/Ssdim5-2*), and the complementation strain *SsDim5-C* exhibited similar nutritional growth and developmental phenotypes on PDA medium. Further statistical analysis indicated no significant differences in growth rates among the strains (Figure 3A,B). Likewise, the number of sclerotium produced and the average weight of each sclerotium by *Ssdim5-1/Ssdim5-2* showed no significant differences compared to the WT and complementation strain on each culture plate (Figure 3C,D). These results imply that *Ss*Dim5 does not impact the mycelial growth and sclerotium formation of the *S. sclerotiorum.*

### 3.4. Deletion of SsDim5 Impairs the Virulence of S. sclerotiorum

To investigate the role of *SsDim5* in pathogenicity, isolated leaves of *B. napus* were inoculated with the WT strain, *SsDim5* knockout strains (*Ssdim5-1/Ssdim5-2*), and the complemented strain (*SsDim5-C*). After 24 h of infection, the lesion areas caused by the knockout strains *Ssdim5-1/Ssdim5-2* were significantly smaller than that induced by the WT, whereas the lesion area caused by the complementation strain *SsDim5-C* resembled that of the WT (Figure 4A,B). Subsequently, each strain was employed to infect detached *N. benthamiana* leaves, and the infection outcomes mirrored those observed in *B. napus* leaves (Appendix A). Notably, the virulence of the knockout strains *Ssdim5-1/Ssdim5-2* was significantly diminished both on *B. napus* and *N. benthamiana* leaves, suggesting that the reduced virulence is not specific to a particular host species.

Appressorium formation and oxalic acid (OA) production are key strategies for regulating *S. sclerotiorum* virulence. Since the deletion of *SsDim5* exhibits a reduced virulence phenotype, the OA production and appressoria of the knockout strain *Ssdim5-1/Ssdim5-2* was tested. The results showed that the morphology and number of appressoria formed by the knockout strain *Ssdim5-1/Ssdim5-2* were similar to those of the WT and complemented strain (*SsDim5-C*), with no significant difference, whether on a glass slide (Figure 4C) or onion epidermal cells (Figure 4D). In addition, by detecting OA production on PDA supplemented with bromophenol blue, as shown in Figure 4E, the knockout strain *Ssdim5-1/Ssdim5-2* had the same phenotype as the WT and complemented strain *SsDim5-C*. These experiments indicate that *Ss*Dim5 plays a critical role in virulence, but not by affecting appressorium formation and OA production.

### 3.5. H3K9 Trimethylation Levels Are Significantly Reduced in SsDim5 Knockout Mutants

To investigate the role of *SsDim5* in histone modification, Western blot analysis was employed to assess the levels of histone methylation in the WT strain, *Ssdim5-1/Ssdim5-2*, and *SsDim5-C*. When probed with an antibody against histone H3, all strains exhibited bands of similar size (Figure 5). However, noteworthy differences emerged when examining trimethylation at H3K9. Abundant specific bands corresponding to trimethylated H3K9 were detected in the WT strain and *SsDim5-C*, while the knockout strains *Ssdim5-1/Ssdim5-2* showed a complete absence of signals for trimethylated H3K9 (Figure 5), indicative of a crucial role of *Ss*Dim5 in H3K9 trimethylation.

### 3.6. SsDim5 Is Related to the Synthesis of Mycotoxins

To further investigate the biological functions of *Ss*Dim5 in *S. sclerotiorum*, whole-genome expression profiling of the WT and *SsDim5* knockout strain (*Ssdim5-1*) mycelia was conducted through RNA sequencing (RNA-Seq). The results revealed 544 differentially expressed genes (DEGs), with 205 upregulated and 339 downregulated (*p*-value < 0.01 and fold change ≥2) (Figure 6A). Subsequently, an enrichment analysis based on Gene Ontology (GO) was performed on the DEGs, resulting in the assignment of 22 GO terms across three categories: biological processes (10 terms), cellular components (3 terms), and molecular functions (8 terms) (Appendix A). Notably, the biological process of “mycotoxins biosynthesis process” was significantly enriched in *Ssdim5-1* (Figure 6B). According to the transcriptional profile, among the 6 genes involved in “mycotoxins biosynthesis process”, *SS1G_13355*, *SS1G_07229*, *SS1G_02251*, *SS1G_13850*, and *SS1G_13358* exhibited significantly lower expression levels in *Ssdim5-1* compared to the WT, while *SS1G_13636* showed significantly increased expression in *Ssdim5-1* (Figure 6C). qRT-PCR validation of these 6 genes in the WT, *Ssdim5-1/Ssdim5-2* and *SsDim5-C* further confirmed the consistent modulation pattern. Specifically, *SS1G_13355*, *SS1G_07229*, *SS1G_02251*, *SS1G_13850*, and *SS1G_13358* were suppressed in *Ssdim5-1* and *Ssdim5-2*, while *SS1G_13636* was enhanced in both *Ssdim5-1* and *Ssdim5-2* (Figure 6D). These findings suggest that *Ss*Dim5 regulates the synthesis of mycotoxins in *S. sclerotiorum*.

### 3.7. Regulation of SsDim5 in Response to Environmental Stresses

The synthesis of mycotoxins is predominantly induced by factors such as oxidative stress, nutritional stress and various external environmental stimuli. To assess the role of *Ss*Dim5 in the resistance of *S. sclerotiorum* to external environmental stimuli, the growth performance of the WT, *Ssdim5-1/Ssdim5-2* knockout strains, and the complemented strain *SsDim5-C* was tested under different stress conditions. The growth inhibition rates of the *Ssdim5-1/Ssdim5-2* knockout strains were significantly higher than those of the WT strain and the complemented strain *SsDim5-C* on PDA plates containing 1M NaCl, 1M KCl, and 1M Sorbitol (Figure 7A,B). This indicates that *SsDim5* is associated with the high osmotic tolerance of *S. sclerotiorum*. Similarly, the growth inhibition rates of the *Ssdim5-1/Ssdim5-2* strains were higher than those of the WT and the complemented strain *SsDim5-C* on PDA plates containing 10 mM H_2_O_2_ and 0.005% SDS, suggesting that *SsDim5* also plays a crucial regulatory role in oxidative stress and cell integrity in *S. sclerotiorum* (Figure 7A,B). Together, these results indicate that *SsDim5* plays a crucial regulatory role in the response of *S. sclerotiorum* to environmental stressors.

## 4. Discussion

In eukaryotes, lysine methylation modification of histones is mediated by histone lysine methyltransferases (KMT) [38], and H3K9 methylation relies on the activity of the KMT1 family [39]. Here, we identified a putative H3K9 methyltransferase, named *Ss*Dim5, by performing blast analysis on the protein sequence of the KMT1 family member *Bc*Dim5 in *B. cinerea. Ss*Dim5 is predicted to have a SET (Su(var)3–9, Enhancer-of-zeste, and Trithorax) domain, commonly associated with histone N-terminal lysine methyltransferase activity [40]. Previous studies have shown that human SUV39H1 and mouse Suv39h1 possess typical SET domains and catalyze directed H3K9 methylation modifications [41]. The Dim5 in *N. crassa* can specifically catalyze trimethylation of H3K9 [27]. In *S. pombe*, the histone methyltransferase Clr4 is responsible for monomethylation, dimethylation, and trimethylation of H3K9 [42]. Moreover, the deletion of *BcDim5* in *B. cinerea* resulted in the loss of H3K9me3 [28]. Similar to previous results, Western blot analysis indicates that H3K9me3 is nearly completely abolished in the *SsDim5* knockout mutant, and restored in the complemented strain, suggesting that *Ss*Dim5 indeed functions as an H3K9 methyltransferase in vivo and affects H3K9me3 in *S. sclerotiorum*. However, some studies have suggested that the disruption of *Dim5* in *Beauveria bassiana* leads to the loss of H3K9me3 and a significant reduction in H3K4me1/me2, H3K9me1/me2, and H3K36me2 [43]. Therefore, it cannot be ruled out that *Ss*Dim5 in *S. sclerotiorum* may simultaneously possess other histone methylation modifications.

At different stages of growth, development, and infection, pathogens undergo extensive transcriptional reprogramming. H3K9 methylation is typically closely associated with heterochromatin formation, influencing various gene expressions by modulating chromosomal accessibility [37]. Through a gene knockout study of *Ss*Dim5 in *S. sclerotiorum*, we discovered an indispensable role for *Ss*Dim5 in the virulence of the fungus, without affecting its hyphal development and sclerotium formation. In the case of *Trichoderma reesei*, the absence of *TrDim5* resulted in impaired vegetative growth and conidiation [44]. The deletion mutant of *FvDim5* in *Fusarium graminearum* (*ΔFvDim5*) exhibited significant defects in conidiation, perithecium formation, and fungal virulence [29]. Similarly, the disruption of *BcDim5* in *B. cinerea* led to a significant reduction in hyphal growth, conidiophore production, and sclerotium yield, accompanied by a decrease in virulence [28]. Undoubtedly, our research results once again confirm the conservation of Dim5 in facilitating the physiological function of pathogen infection. However, there are also differences existed in Dim5 function among different species. In *S. sclerotiorum*, Dim5 is expressed at various stages of growth and development, yet no significant growth inhibition was observed through the observation of knockout mutants. Therefore, we speculate that *Ss*Dim5 may have redundant functions in the regulation of growth and development processes in *S. sclerotiorum*.

Genome sequencing has revealed that gene clusters involved in fungal secondary metabolism are often located near telomeres or heterochromatin regions [45]. H3K9 methylation, closely associated with the establishment of heterochromatin [46], has been demonstrated as an effective regulatory mechanism for disrupting or maintaining heterochromatin, impacting the generation of fungal secondary metabolites [47,48]. Here, RNA-seq analysis unveiled dysregulation in the expression of metabolism-related genes in the *SsDim5* knockout mutant, with significant enrichment observed in the pathway of mycotoxins biosynthesis through Gene Ontology (GO) analysis. Secondary metabolites (SM) play a crucial role in the virulence, development, and overall lifestyle of fungal pathogens. In the interaction between fungi and plant hosts, these metabolites may function as effectors while infecting the host and when recognized by the plant host during the infection process [49]. Mycotoxins biosynthesis is a subset of secondary metabolites often overproduced in response to external stressors. The main factors that enhance mycotoxin production include oxidative stress, nutritional stress, light stress, as well as environmental factors such as pH, temperature, water activity, fungicides, and plant secondary metabolites [50]. Studies have demonstrated that fungi regulate oxidative bursts through mycotoxins, enhancing ecological benefits [51]. qRT-PCR results confirmed a significant decrease in the expression of genes related to mycotoxins biosynthesis in *SsDim5* knockout mutant. Additionally, the *SsDim5* knockout strain exhibited increased sensitivity to osmotic stress, oxidative stress, and substances damaging cell integrity. Intriguingly, disturbances in the redox state of *S. sclerotiorum* affected the accumulation of OA [52], impacting the levels of osmotic stress, high salt, and cell wall stress-related functional genes that influence virulence [53,54,55,56,57]. Therefore, it can be inferred that in *S. sclerotiorum*, the normal function of Dim5 is crucial for the synthesis of fungal secondary metabolites, especially mycotoxins, directly or indirectly influencing the virulence of *S. sclerotiorum.*

## 5. Conclusions

In summary, our results indicate that *Ss*Dim5 possesses histone H3 lysine methyltransferase activity, and plays a crucial regulatory role in the pathogenicity of *S. sclerotiorum* through the regulation of fungal mycotoxins biosynthesis, and the response to external stressors. This study therefore provides theoretical guidance for the development of new target points for the prevention and control of Sclerotinia stem rot.

## Figures and Tables

**Figure 1 jof-10-00271-f001:**
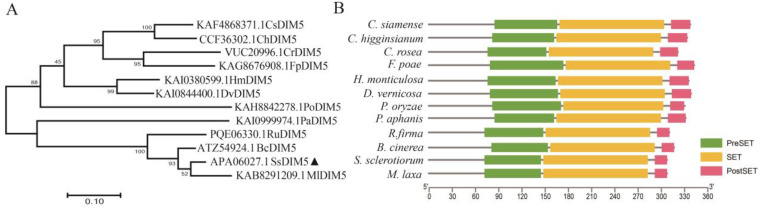
Phylogenetic tree, conserved protein domain of *SsDim5*. (**A**) Phylogenetic analysis of *Ss*Dim5 was conducted using the neighbour-joining method, aligning protein sequences with ClustalW based on JTT in MEGAX. Statistical confidence in the phylogenetic relationships was evaluated through 1000 bootstrap replicates. *Ss*Dim5 was marked with a black triangle, and UniProt database entry numbers were provided in brackets. (**B**) Conserved protein domains of *Ss*Dim5 and its orthologues were predicted using TBtool.

**Figure 2 jof-10-00271-f002:**
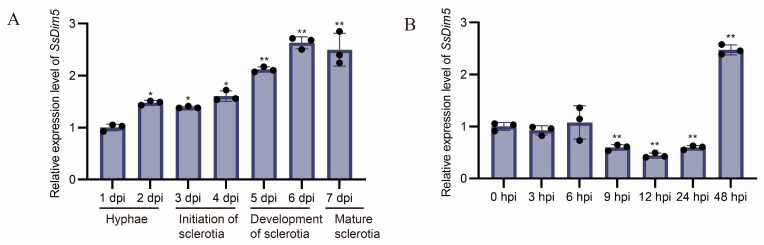
Expression analysis of *SsDim5*. (**A**) Expression levels of *SsDim5* at different growth stages of *S. sclerotiorum*. Utilizing *SsTubulin1* as the reference gene, average values and standard deviations were computed based on data from three independent biological replicates. Differences were evaluated using the one-way ANOVA test. * denotes *p* < 0.05, ** denotes *p* < 0.01. (**B**) Expression levels of *SsDim5* at different infection stages in *B. napus* leaves. Utilizing *SsTubulin1* as the reference gene, average values and standard deviations were computed based on data from three independent biological replicates. Differences were evaluated using the one-way ANOVA test. ** denotes *p* < 0.01.

**Figure 3 jof-10-00271-f003:**
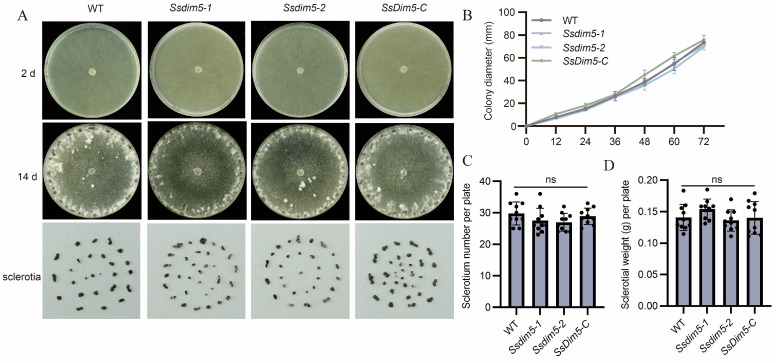
Phenotypes of *SsDim5* knockout and complemented strains. (**A**) Phenotypes of the WT, *SsDim5-1*, *SsDim5-2*, and *SsDim5-C* at 2 or 14 days. One representative biological replicate was shown. (**B**) Radial growth length assessment on PDA of WT, *Ssdim5-1*, *Ssdim5-2*, and *SsDim5-C* strains. Average values and standard deviations were computed based on data from three independent biological replicates. (**C**) Number of sclerotia per plate. Average values and standard deviations were computed based on data from ten independent biological replicates. Differences were evaluated using the one-way ANOVA test. (**D**) Sclerotial weight per plate. Average values and standard deviations were computed based on data from ten independent biological replicates. Differences were evaluated using the one-way ANOVA test.

**Figure 4 jof-10-00271-f004:**
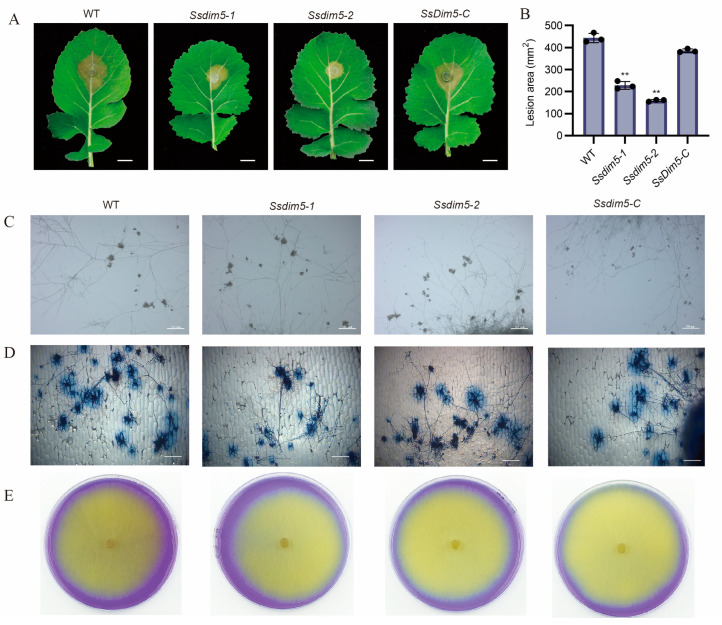
Pathogenicity assays of individual strains. (**A**) Disease phenotype of WT, *Ssdim5-1*, *Ssdim5-2* and *SsDim5-C* leaves of *B. napus*. Photographs were taken at 24 hpi. One representative biological replicate was shown. Bar = 1 cm. (**B**) Statistical analysis of the lesion area in panels. WT, *Ssdim5-1*, *Ssdim5-2* and *SsDim5-C*. Average values and standard deviations were computed based on data from three independent biological replicates. Differences were evaluated using the one-way ANOVA test. ** denotes *p* < 0.01. (**C**) The morphology of the appressoria of each strain after 16 h of inoculation on a glass slide under a stereomicroscope. Bar = 500 μm. (**D**) The morphology of the appressoria of each strain after 16 h of inoculation on the onion epidermis under a stereomicroscope. Bar = 200 μm. (**E**) Mycelium of WT, *SsDim5-1*, *Ssdim5-2* and *Ssdim5-C* strains cultivated on PDA medium supplemented with bromophenol blue.

**Figure 5 jof-10-00271-f005:**
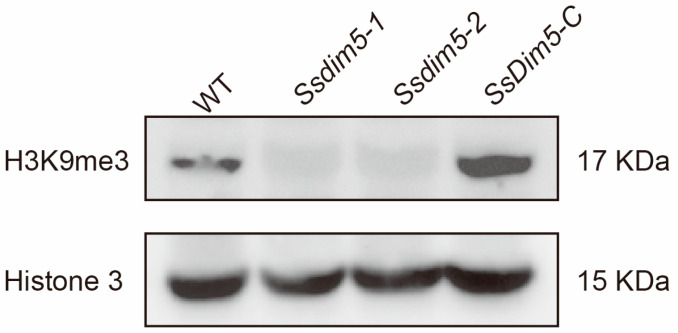
Western-blot analysis of *S. sclerotiorum* using antibodies histone H3 or trimethylated H3K9.

**Figure 6 jof-10-00271-f006:**
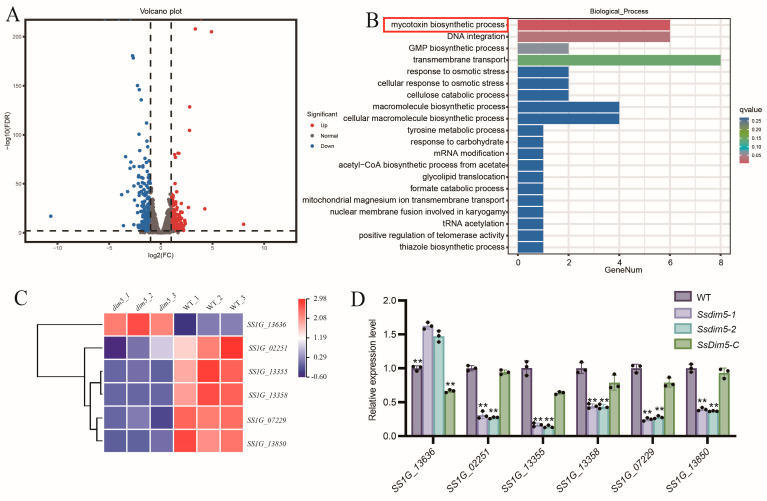
RNA-Seq analysis of WT and *Ss*Dim5 knockout strains. (**A**) The volcano plot of differentially expressed genes (DEGs) (*SsDim5* knockout strains vs. WT). (**B**) Significantly enriched Gene Ontology (GO) terms for all DEGs. (**C**) The expression profiles of mycotoxins biosynthesis genes. (**D**) The expression levels of mycotoxins biosynthesis genes by qRT-PCR. Utilizing *SsTubulin1* as the reference gene, average values and standard deviations were computed based on data from three independent biological replicates. Differences were evaluated using the one-way ANOVA test. ** denotes *p* < 0.01.

**Figure 7 jof-10-00271-f007:**
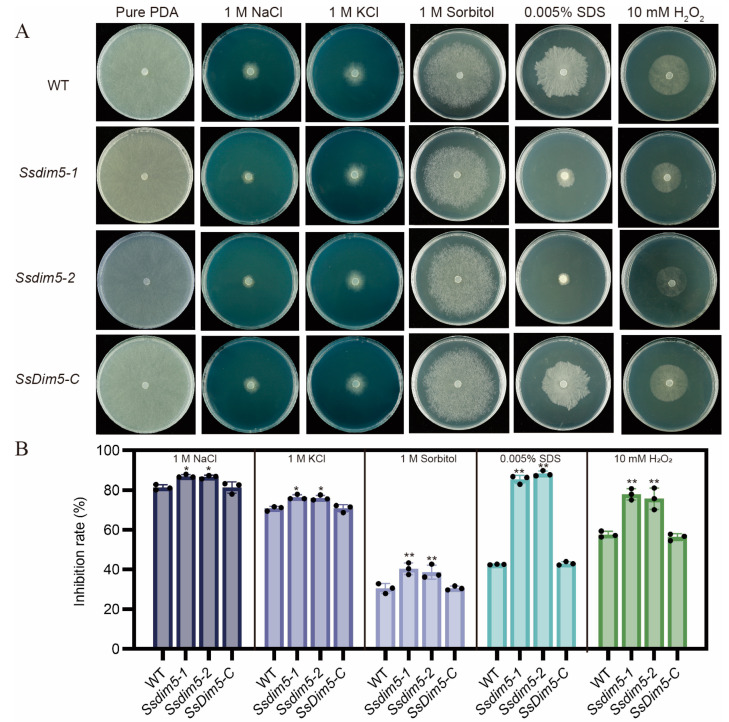
The growth of individual strains in the presence of various stressors. (**A**) Phenotypes of the WT, *Ssdim5-1*, *Ssdim5-2* and *SsDim5-C* supplemented with 1 M NaCl, 1 M KCl, 1 M Sorbitol, 0.005% SDS, 10 mM H_2_O_2_, respectively. (**B**) Statistics of inhibition rates under various stresses. Average values and standard deviations were computed based on data from three independent biological replicates. Differences were evaluated using the one-way ANOVA test. * denotes *p* < 0.05, ** denotes *p* < 0.01.

## Data Availability

The data presented in this study are available on request from the corresponding author.

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
