# Peer review of "Histone Methyltransferase SsDim5 Regulates Fungal Virulence through H3K9 Trimethylation in Sclerotinia sclerotiorum"

_jof, 2024, doi:10.3390/jof10040271_

Round 1

Reviewer 1 Report

The manuscript by Qin et al. is a good work that revolves around the Histone methyltransferase SsDim5. In this manuscript, the authors knocked out SsDim5 and conducted various experiments to establish its association with traits such as fungal growth rate, virulence, and mycotoxin level. Overall, it is an excellent manuscript, and I can only identify a few points that require minor improvement. As a result, I recommend accepting the manuscript after a minor revision.

Line 32-35: The sentence is miswritten and should be rewritten. Ex: The cutinase enzyme, produced by the fungus, can inhibit host salicylic acid synthesis, thereby promoting infection [5].   Line 84:  our laboratory stock- be specific   Line 91-92: The bootstrap number should be mentioned here as well, in addition to Figure 1.   Line 115-116: The authors should mention how they extracted fungal DNA at different time points, as explained in Figure 1C & D.   Figure 2A: labels for top, middle, and bottom    - It would be even great if the authors could provide RNA-Seq data of the plant side when inoculated with WT vs mutant to see what genes are up-regulated in the host plant. But it is optional.   

Author Response

We thank the reviewers for their valuable comments, which certainly help us to improve the quality of our manuscript. All comments are addressed on a point-by-point basis below, where letters C & R denote Comment and Response, respectively. The modifications of the text, figures and tables have been highlighted.

Reviewer 2 Report

Overall Summary: Qin et al. present a manuscript that evaluates the impact of a histone methyltransferase SsDim5 gene on virulence and transcriptomic changes in Sclerotinia. While the authors do employ several approaches to study the effects of this gene on virulence and other adaptive factors, there is a lack of detail in terms of the methodology description, results presented and discussion of results. There is also a need to critically consider the results and their physiological relevance. Specific comments are listed below. 

Specific Comments:

·         Figure 1B: If the domain profiles correlate directly with the branches of the phylogenetic tree, it needs to be mentioned clearly. Also, what are the actual annotations of the domains on these proteins. The figure only represents them as a group of amino acids that have been divided into groups, however it would be more useful to know the actual annotation of the domains present in these proteins.

·         Figure 1C: There is no zero timepoint sample present in this dataset. This should be included as well. Also, since the growth of the hyphae will be exponential over time, it is unclear how the samples were collected and how they were normalized for the biomass. Also, what was the process of collecting hyphae to keep the sampling constant in terms of the location on the plate, etc. These points need to be mentioned in detail.

·         Figure 1D: There needs to be more explanation about why specific time points were chosen for sampling. Also were samples taken multiple days post inoculation? These time points are relatively close. How does this relate to disease progression in the host, which could occur on a slower timescale. Also, what was the process of sampling, did RNA extraction occur from the entire leaf tissue? There is a general lack of detail about the procedure, which should be improved.

·         Figure 2: Are the two different mutants evaluated just two different colonies of the same mutation processes? If so, which one of these is being used for complementation. This is unclear. Ideally, both mutant strains should be complemented, and these should be included in the study.

·         Figure 5A: The volcano plot should have some of the annotations for DEGs highlighted on the graph.

·         In addition to figure 5, the manuscript should include a list of top 20-50 positively and negatively regulated DEGs along with the p-values and the Log2 fold change in expression. This should be discussed in detail with specific top hits being described.

·         Figure 6: While statistics show that NaCL, KCL and Sorbitol significantly reduced mycelial growth, the effective inhibition is almost negligible, it is unclear how significant this result is physiologically. Other than SDS and H2O2, there isn’t a clear distinction between the WT and mutant. This needs to be explained in the results. There should be a correlation made between just minor statistical differences and any true physiological differences.

·         What is the time course of the experiment conducted in Figure 6? If the authors claim there is a physiological difference between the mutant and WT for certain conditions, a longer time course should be conducted along with timepoints. Currently this is not evident.

Author Response

(The authors gave the same response as above.)

Reviewer 3 Report

The authors characterized the histone methyltransferase SsDim5 in Sclerotinia sclerotiorum, examining its role in histone H3 lysine trimethylation, regulation of mycotoxin synthesis, pathogenicity and response to external stresses. They found that deletion of SsDim5 leads to reduced virulence in Ssdim5-1/Ssdim5-2 mutants, accompanied by a significant reduction in H3K9 trimethylation. Transcriptome analysis showed reduced expression of genes related to mycotoxin biosynthesis in mutants with deleted SsDim5. In addition, the absence of SsDim5 affects the response of the fungus to oxidative and osmotic stresses, as well as cell integrity. This work is interesting and certainly important, but it has certain drawbacks, which I list below.

1.     In the Introduction section, it is worthwhile to give a brief overview of the chromatin tags relevant to the work and to note that H3K9 is a constitutive heterochromatin tag, as there is also facultative heterochromatin. In this context, it is also worthwhile to take a slightly broader look at the processes for which H3K9 trimethylation is required, beyond pathogenesis and toxin synthesis.

22.      In order to better determine the genotypes of the Ssdim5-1, Ssdim5-2 and SsDim5-C strains, it would be reasonable to perform Sanger sequencing of the corresponding genome region and attach the results as Supplementary Files.

33.      Line 106. “Agrobacterium tumefaciens” should be written in italics.

44.      It is necessary to indicate which accession numbers in the NCBI database the B. cinerea and S. sclerotiorum genomes used in this study have and to note which database the designations SS1G_01550 SS1G_13355, SS1G_07229, SS1G_02251, SS1G_13850 and SS1G_13358 refer to so that any reader can work with these sequences. The accession number of the SsTubulin gene used as a reference gene should also be specified.

55.      In the text of Section 2.5. “Quantitative Real-Time PCR (qRT-PCR) analysis”, reference should be made to Table S1 containing the primer sequences.

66.      In Section 2.10, "RNA sequencing and data analysis", the programs used at all stages, including their versions, should be specified in as much detail as possible. It should be stated that fastqc or other software for read quality assessment, trimmomatic, cutadapt, etc. for further processing, STAR, HISAT2, kallisto, etc. for alignment or pseudo-alignment to the genome or transcriptome, the version of DESeq2 and the software used for functional analysis (GO and KEGG) were used.

77.      The graphs (Figure 1C and Figure 1D) should be labelled with what is happening to the fungus at a given time (i.e. when the stages of sclerotia formation and maturation are occurring, different stages of infection, etc. to correlate SsDim5 gene expression with the stage of the fungal life cycle).

88.      In Figure 2A, either on the figure itself or in its legend, it should be signed what is shown in the top, middle, and bottom photo for each strain.

99.      It should be briefly explained what appressoria are, what role oxalic acid plays in the life of fungus and how it is related to virulence.

110.  It should be clarified which mycotoxins are present in S. sclerotiorum and the biosynthesis of which of them is discussed in section 3.6. "SsDim5 is associated with mycotoxin synthesis". Do the authors have experimental data on the content of these mycotoxins in WT compared to Ssdim5-1, Ssdim5-2 and SsDim5-C strains?

 11. For the differentially expressed genes SS1G_13355, SS1G_07229, SS1G_02251, SS1G_13850 and SS1G_13358, it should be written which proteins they encode, if this is unknown, an attempt should be made to predict by BlastP or by their domains, and it should be noted whether they are enzymes or regulatory proteins.

1 12. Figure 5A and Figure 5B should be larger or higher resolution as it is currently very difficult to see the figure captions.

1  13. It is worthwhile to extend the transcriptome analysis. A heatmap of clustered differentially expressed genes should be added, for example, as on the Figure 2 in this article: https://www.frontiersin.org/journals/physiology/articles/10.3389/fphys.2020.01107/full.

1  14. It is also necessary to deepen the functional analysis by plotting 4 analogous graphs: (1) GO enrichment for upregulated genes, (2) GO enrichment for downregulated genes, (3) KEGG enrichment for upregulated genes, (4) KEGG enrichment for downregulated genes. For example, as in Figure 2 in this paper: https://www.hindawi.com/journals/jcpt/2023/7069469/. It will then become clearer which processes and to what extent are affected by the SsDim5 disfunction.

1  15. Among the differentially expressed genes one should look for genes encoding effector proteins and others proteins important for pathogenesis. If they are downregulated, this will further explain the reduced virulence of mutant strains.

Author Response

(The authors gave the same response as above.)

Reviewer 4 Report

The manuscript "Histone methyltransferase SsDim5 regulates fungal virulence through H3K9 trimethylation in Sclerotinia sclerotiorum" provides valuable information on one of the main mechanisms of epigenetic regulation, which could play a relevant role in the control of gene expression. The approach used with the elimination of the Dim5 gene, the transcriptomic studies, infection, evaluation of the effect of different stresses and methylation provides results that indicate that the H3K9 methyltransferase SsDim5 is relevant for the trimethylation of H3K9, which also regulates the virulence of fungi on host plants by impacting mycotoxin biosynthesis and environmental stress response in S. sclerotiorum. It is therefore a very complete work and an in-depth study.

The only recommendations I would make are that figure 1 be divided into two, separating the phylogenetic analysis from the expression analysis and the second figure be placed after section 3.2 of results.

In Figure 5 I wanted to improve the quality of section B and made mention both in results and in discussion of the fact that in Significantly enriched Gene Ontology (GO) terms for all DEGs transmembrane transport stands out. Nothing is commented on and it is striking.

It would also be necessary to clarify why in the toxin biosynthesis pathway, there is a gene that is induced and the emas are repressed and what effect this has on toxin synthesis.

Author Response

(The authors gave the same response as above.)

Round 2

Reviewer 2 Report

Authors addressed the comments. 

Authors addressed the comments. 

Reviewer 3 Report

The authors have substantially improved the manuscript in accordance with the reviewers' comments. I have no further significant comments and propose to accept the manuscript for publication.

Various technical points in the manuscript can be corrected at the proofreading stage.